# Breeding range shift of the red-crowned crane (*Grus japonensis*) under climate change

Liwei Liu[1☯], Jishan Liao[2☯], Yongbo Wu[3], Yinlong Zhang[1]*

**1** Collaborative Innovation Center of Sustainable Forestry in Southern China of Jiangsu Province, Nanjing Forestry University, Nanjing, P.R. China, **2** Faculty of Forestry, University of Toronto, Toronto, ON, Canada, **3** College of Biology and the Environment, Nanjing Forestry University, Nanjing, Jiang Su, P.R. China

☯ These authors contributed equally to this work.
* ecoenvylz@163.com

**Data Availability Statement:** All relevant data are within the manuscript and its Supporting Information files.

**Funding:** This project was supported by National Key R&D Program of China (2016YFC0502704) and Priority Academic Program Development of

## Abstract

The red-crowned crane (*Grus japonensis*) is an endangered species listed by International Union for Conservation of Nature (IUCN) HARRIS J (2013). The largest population of this species is distributed mainly in China and Russia, which is called continental population SU L (2012)–Curt D (1996). This population is migratory, which migrates from its breeding range located in Northeast China and Southern Russia, to the wintering range in the south of China to spend the winter every year. The breeding range of this species is critical for red-crowned crane to survive and maintain its population. Previous studies showed the negative effects of habitat loss and degradation on the breeding area of red-crowned crane Ma Z (1998), Claire M (2019). Climate change may also threat the survival of this endangered species. Previous studies investigated the impacts of climate change on the breeding range or wintering range in China Wu (2012), [1]. However, no study was conducted to assess the potential impacts of climate change on the whole breeding range of this species. Here, we used bioclimatic niche modeling to predict the potential breeding range of red-crowned crane under current climate conditions and project onto future climate change scenarios. Our results show that the breeding range of the continental population of red-crowned crane will shift northward over this century and lose almost all of its current actual breeding range. The climate change will also change the country owning the largest portion of breeding range from China to Russia, suggesting that Russia should take more responsibility to preserve this endangered species in the future.

## Introduction

The red-crowned crane (*Grus japonensis*) is among the largest and rarest cranes in the world [2]. It has great cultural value in China as it is the symbol of luck, longevity and fidelity. Historically, this species was distributed widely in the eastern Asia. However, the range of this species contracted and its population declined greatly over last centuries, reaching lowest number in the 1950s [3,4]. Although the number of red-crowned cranes is increasing in the recent years due to great conservation efforts, the population size of this endangered species is still low [5,3]. Red-crowned crane has been listed as first-class national key protected wild animal in China [6]. It is also listed as an endangered species by IUCN [7]. This species is distributed

Jiangsu Higher Education Institutions (PAPD). There was no additional external funding received for this study.

**Competing interests:** The authors have declared that no competing interests exist.

mainly in five countries: Russia, Mongolia, China, Korea and Japan. The populations in Russia, Mongolia, China and Korea are migratory, while the population in Japan is non-migratory because abundant food is available [8,9]. The entire migratory population of the red-crowned crane, which is also called continental population, breeds in the Siberia (eastern Russia), northeastern China and northeastern Mongolia (Amur-Heilong River basin) in the spring and summer, from March to September. The largest portion of the continental population inhabits the lower and middle part of Amur-Heilong River basin in China [7,6,2]. The remaining continental population breeds in grass marshlands of the Zeya-Bureya and Middle Amur-Heilong plain, Ussuri River valley, Khanka Lake in the Russian part of the Amur-Heilong River basin [3]. In the autumn, the continental population of red-crowned crane migrates to its wintering range, such as Korea and east-central China, to spend the winter [10].

The anthropogenic habitat change, such as development of agriculture, urbanization, and conversion of wetland to human-built area, and increasing of captive red-crowned cranes harvested from the wild in China, have caused a significant decline of the population and loss of habitats of the red-crowned crane [2,5,3,4,11]. As a result, the breeding and wintering ranges of red-crowned crane has been shrunk to a relatively small range (Fig 1). This species prefers to nesting in wetlands and river floodplains [2,12,13]. The conversion of wetlands to agriculture lands and built areas threats the nesting area of the red-crowned crane [3]. In the wintering range, the red-crowned cranes prefer to inhabit in paddy fields, grassy tidal flats, mudflats and rivers, which means that it has very strict habitat preferences [6,4,14]. Therefore, this species is prone to suffer risk from habitat modification and loss resulting from anthropogenic environmental change. In addition to threats from human-induced habitat modification and loss, climate change may further threat endangered species [15–19]. Some studies have already showed that birds track their niches by moving along climate change [20–23]. Climate change may shift both breeding and wintering ranges of the red-crowned crane out of its current ranges. The possible contraction of breeding and wintering ranges as a result of climate change, and the incapability of the red-crowned crane to track its climatic niche, could threaten future survival of this endangered bird species [24].

It is of societal importance to predict the potential shift of the red-crowned crane's breeding range in the face of climate change. Climate change may cause the loss of effectiveness of the current protected area made mainly for this endangered species [25], such as Zhalong National Natural Reserve and Yancheng Biosphere Reserve. In this study, we only consider the breeding range of the continental migratory populations. Policy makers do not necessarily consider the potential effects of climate change on the ranges of endangered species when planning the protection areas for them [26–29]. The populations of red-crowned crane may move out of the current protected area designed for it when climate changes, which jeopardizes the effectiveness of the protected area. Because the breeding range of continental population of the red-crowned crane is distributed along the border of China, Russia and Mongolia, climate change may also cause some populations move northward to Russia and Mongolia, which thereby changes the role of each country for preserving this endangered species. Therefore, predicting range shifts of red-crowned crane under changing climate will contribute greatly to policy making process regarding preservation of this endangered species.

Previous studies used ecological niche modeling, which relates the occurrence of a species to environmental variables considered to determine the distribution of this species, to predict the impacts of climate change on global biodiversity of birds [30–32]. Ecological niche modeling was also used to predict the geographic distribution of the red-crowned crane under climate change [33,34]. However, these studies focused mainly on a single country. For example, Wenjun et al. used Maximum Entropy (MaxEnt) [35], a powerful and popular ecological niche modeling technique, and a variety of variables to predict the effects of climate change on

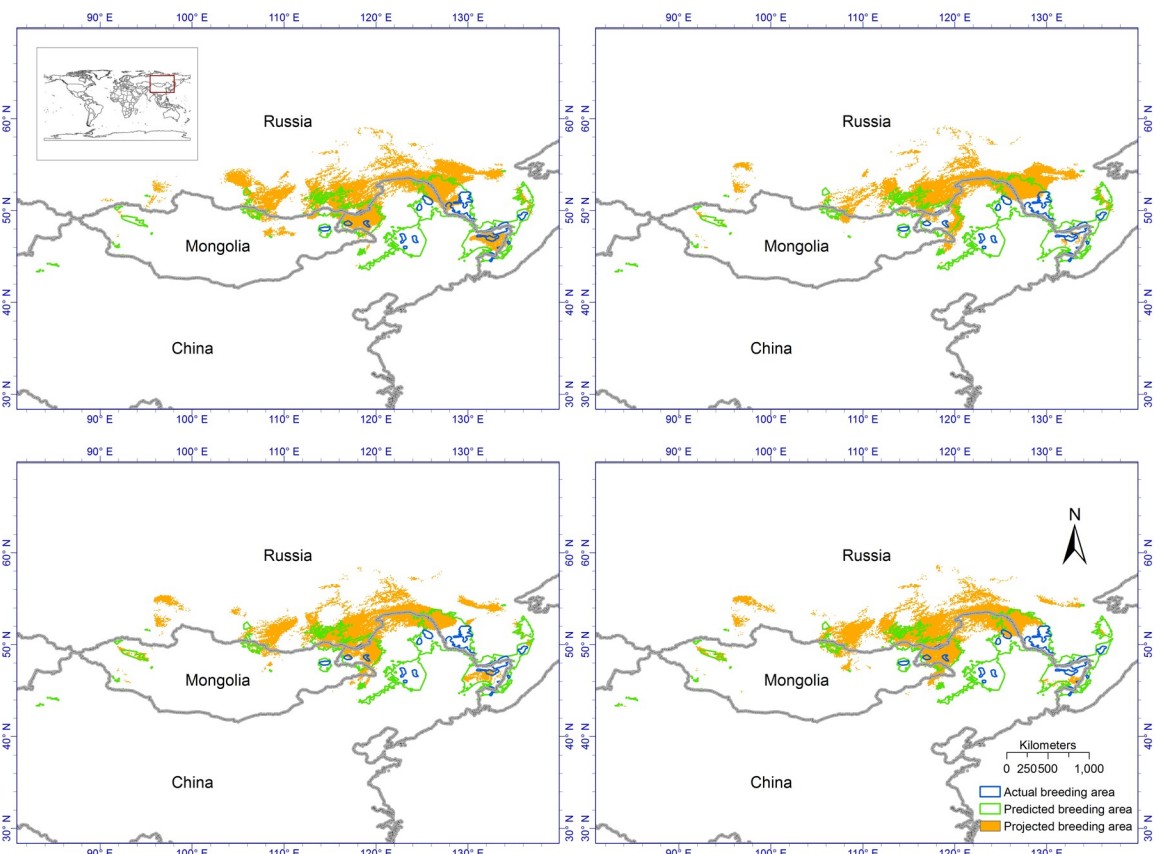

**Fig 1. Predicted breeding range of the continental population of the red-crowned crane at 2030s, 2050s, 2070s and 2080s** [标注到图上]. Blue polygons are the current actual breeding area, green polygons are the predicted potential breeding area under current climate condition, and orange areas are the predicted potential breeding area climate change. Scenario is RCP2.6, low emission scenario.

the breeding range of this species in China [33]. It predicted the contraction of the breeding range of red-crowned crane in China. But it failed to provide information on whether Russia or Mongolia gain some potential suitable habitats under changing climate. This limitation makes these studies incapable of providing information to guide the policy making for conserving this important endangered species, especially in the international collaboration setting. Here, we used Maxent [35–37], to predict the shift of the breeding range of the continental migratory population of the red-crowned crane under changing climate.

## Methods

The breeding range of the continental population of the red-crowned crane is distributed in the Siberia (eastern Russia), northeastern China and northeastern Mongolia (Amur-Heilong River basin) (Fig 1). However, there is no adequate occurrence point data available at this broad scale for building bioclimatic niche models. We obtained the extent of occurrence polygons of continental population of Red-crowned crane from AMUR-HEILONG RIVER BASIN Information Center (http://amur-heilong.net/http/03_species/0309Redcrowned_crane.html). We digitized and georeferenced those polygons using ArcGIS 10.2 [38]. We then converted the occurrence polygon data to lattice data to generate the pseudo-occurrence points of red-crowned crane, because Maxent can only use point data. Each point represents a grid of 5*5

arc-minutes (~10 km spatial resolution). Andriamasimanana and Cameron also used this kind of data to predict the impacts of climate change on threatened birds in Madagascar [39].

There are many environmental factors that determine the geographic distribution of the red-crowned crane, such as food resource, availability of nesting site, vegetation cover, and climate [2]. However, the spatial data of most factors are not available at this large scale in our study. Here we only studied the bioclimatic niche of this species using bioclimatic niche modeling. Coetzee et al. used bioclimatic niche modeling to predict range changes in relation to the Important Bird Areas (IBAs) network in southern Africa under climate change [40]. We also attempted to assess the effects of the change of bioclimatic conditions under changing climate on the breeding range of this endangered species. The 19 bioclimatic variables used for bioclimatic niche modeling were obtained from WorldClim dataset (http://www.worldclim. org/) [41]. Bioclimatic variables represent the extreme and general trend of monthly temperature and precipitation (Table 1). This dataset is widely used in predicting distributions of species. The future simulation data of bioclimatic variables was obtained from CCAFS GCM DATA PORTAL (http://www.ccafs-climate.org/data/). It is part of the International Centre for Tropical Agriculture (CIAT) and The CGIAR Research Program on Climate Change, Agriculture and Food Security (CCAFS) [42]. We chose simulations from General Circulation Models (GCMs) used in IPCC 5th report. For emission scenarios, we used all four representative concentration pathways (RCPs), i.e. RCP 2.6 (low emissions), RCP 4.5 (intermediate emissions), RCP 6 (intermediate emissions), and RCP 8.5 (high emissions). Those four RCPs are named based on how much heating they would produce in the year 2100–2.6, 4.5, 6.0, and 8.5 watts per square meter (W/m$^2$), respectively [43]. The monthly maximum temperature, monthly minimum temperature and monthly total precipitation from GCMs were statistically downscaled based on delta method. The 19 bioclimatic variables were then calculated from temperature and precipitation variables (http://ccafs-climate.org/downloads/docs/Downscaling-WP-01.pdf). For climate model, we chose HadGEM2-ES (Hadley Global Environment Model 2—

**Table 1. Bioclimatic variables.**

| Abbreviation | Description |
|---|---|
| Bio_1 | Annual Mean Temperature |
| Bio_2 | Mean Diurnal Range (Mean of monthly (max temp—min temp)) |
| Bio_3 | Isothermality (BIO2/BIO7) (* 100) |
| Bio_4 | Temperature Seasonality (standard deviation *100) |
| Bio_5 | Max Temperature of Warmest Month |
| Bio_6 | Min Temperature of Coldest Month |
| Bio_7 | Temperature Annual Range (BIO5-BIO6) |
| Bio_8 | Mean Temperature of Wettest Quarter |
| Bio_9 | Mean Temperature of Driest Quarter |
| Bio_10 | Mean Temperature of Warmest Quarter |
| Bio_11 | Mean Temperature of Coldest Quarter |
| Bio_12 | Annual Precipitation |
| Bio_13 | Precipitation of Wettest Month |
| Bio_14 | Precipitation of Driest Month |
| Bio_15 | Precipitation Seasonality (Coefficient of Variation) |
| Bio_16 | Precipitation of Wettest Quarter |
| Bio_17 | Precipitation of Driest Quarter |
| Bio_18 | Precipitation of Warmest Quarter |
| Bio_19 | Precipitation of Coldest Quarter |

Earth System), which accounted for more climate change processes and included dynamic vegetation [44]. The spatial resolution of both current and future climate data is 5*5 arc-minutes. The model output from HadGEM2-ES is average on 30 years centered at 2030s (2020–2049), 2050s (2040–2069), 2070s (2060–2089), and 2080s (2070–2099) before downscaling.

We used Maximum Entropy (MaxEnt) [35], which is one of the most popular and powerful modeling technique for predicting distribution of species [45]. For parameter settings, we used default parameter in MaxEnt. We used 10-fold cross validation to assess model performance. Data are divided randomly into 10 subsets with equal sample size. Then nine subsets of the data were used to train model, and the remaining one subset was used to test model performance. We used area under the receiver operating characteristic (ROC) curve (AUC) to assess our model [46]. The AUC value ranges from 0 to 1. Higher AUC value means greater discriminant ability of bioclimatic niche model. Model evaluation was performed on the average AUC of the 10 cross validation runs. For each cross validation run, we chose the threshold which maximize the sum of sensitivity and specificity to convert the predicted suitability binary map (presence and absence). This threshold is equivalent to that from maximizing true skill statistic (TSS), which is recommended as a good measure for the performance of ecological niche model when predictions are binary maps [47]. We summed together the 10 binary maps from 10 fold cross validation models. We then set the areas where the grid value larger than 5 (i.e. areas where at least 6 models predicted the presence of this species) as the presence area for the continental population of the red-crowned crane.

Variable importance measures the relative importance of environmental variables in determining the distribution of species in a niche model. Two measurements of variable importance can be calculated in MaxEnt software: percent contribution and permutation importance [37]. Percent contribution depends on the particular path to get to the optimal solution and a different algorithm could result in different percent contribution values, therefore we did not use it in this study. Permutation importance measure depends only on the final MaxEnt model [37]. The importance for each variable is determined by randomly permuting the values of that variable in the training dataset and measuring the resulting decrease in training AUC, and normalized to percentages. A larger decrease indicates that that variable is more important in the model. The ranking of relative importance can help us find out which variables are more important in affecting the distribution of the red-crowned crane, and assessing the potential effects of changing of those variables under climate change on the breeding range of the red-crowned crane.

To assess the effects of human activities in breeding range of the continental population of the red-crowned crane, we used human influence index as an indicator of intensity of human activities. The human influence index was calculated from nine global data layers including human population pressure (population density), human land use and infrastructure (built-up areas, nighttime lights, land use/land cover), and human access (coastlines, roads, railroads, navigable rivers). The dataset is produced by the Wildlife Conservation Society (WCS) and the Columbia University Center for International Earth Science Information Network (CIESIN) (available at http://sedac.ciesin.columbia.edu/data/set/wildareas-v2-human-influence-index-geographic). The value of human influence index ranges from 0 to 64, higher value means heavier human activities in this raster cell.

## Results

The average test AUC for the 10 fold cross validation runs is 0.943, and the standard deviation is 0.004. The high AUC value means our models can predict the distribution of the breeding range of the continental population of the red-crowned crane very well. The low standard

deviation means the performance of 10 models from 10 fold cross validation is consistent. The predicted current breeding range area is much larger than the actual breeding range of the continental population of the red-crowned crane, suggesting that other factors such as food resource, nesting preference and human built area restrict the breeding range of this species (Fig 1). It is interesting that there are also some suitable breeding areas in the western China and western Mongolia, where is far from the actual breeding area.

The overall spatial patterns of the changes of projected breeding area under all four emission scenarios are very consistent under changing climate (Figs 1 and S1, S2 and S3). The breeding range of the continental population of the red-crowned crane will expand northward when project onto future climate change scenarios from our bioclimatic niche modeling. The predicted potential breeding range from our bioclimatic niche model would be contracted as climate change (Figs 1 and S1, S2 and S3). The current actual breeding area is mainly distributed in the center part of Northwest China and along the national border shared by eastern part of Northwest China and Southern Russia. As a result of climate change, the breeding area of continental population would shift to the region along the national border shared by northern part of Northwest China and Southern Russia (Figs 1 and S1, S2 and S3). The migratory population of the red-crowned crane, therefore, needs to move northward to track its bioclimatic niche. There are still some differences among projections of four emission scenarios. The major difference among predictions under the four emission scenarios is that the predictions of 2030s, 2050s and 2080s under low emission scenario (RCP2.6) are relatively stable. In addition, the predictions of 2030s, 2050s, 2070s and 2080s under two intermediate emission scenarios (RCP4.5 and RCP 6.0) and high emission scenario (RCP8.5) show that the breeding area would move farther north. We only show RCP2.6, the low emission scenario, in the main text (Fig 1).

The latitudinal pattern of projected breeding area of the red-crowned crane by calculating the number of grid cells in 0.5 arc-degree band demonstrated that the breeding area of the continental population of red-crowned crane shift northward under all four RCPs (Fig 2). The current actual breeding area of the red-crowned crane distributes from 43 arc-degree to 54 arc-degree and peaks around 50 arc-degree (Fig 2A). Under the low emission scenario— RCP2.6, the projected breeding areas in 2030s, 2050s, 2070s and 2080s are very similar and peak at 52 arc-degree, suggesting that the breeding area would be stable after 2030s from predictions of our bioclimatic niche model (Fig 2A). The latitudinal patterns of the projected breeding area under RCP4.5 for 2050s, 2070s and 2080s are similar but the size of breeding area would shrink gradually (Fig 2B). The breeding area at 2080s under RCP6.0 is peaking at 55 arc-degree. There are two peaks of breeding area at 2080s under RCP8.5, i.e. 52 degree and 56 degree. The accelerating climate change under higher emission scenario will cause the breeding area of red-crowned crane move farther north.

The total number of grid cells predicted to be breeding area of the continental population of the red-crowned crane under RCP2.6 increases at 2030s, and decreases sharply at 2050s, and then increase slightly (Fig 3A). The total number of grid cells under the two intermediate emission scenarios–RCP4.5 and RCP6 maintains the similar pattern of change, it increase at the first, and then decreases afterward (Fig 3B and 3C). The grid number of the predicted breeding area under the high emission scenario—RCP8.5, decreases quickly after 2050s to only half of the number under current climate condition (Fig 3D). The number of predicted grid cells of breeding area of the continental population of the red-crowned crane in both China and Mongolia decreases generally as climate changes over the time periods (Fig 3). For Russia, the number of grid cells predicted to be breeding area for the red-crowned crane under RCP4.5, RCP6.0 and RCP8.5 increases at first and then decreases under changing climate. The number of grid cells in Russia at the end of this century under RCP 2.6, RCP4.5 and RCP6.0 is

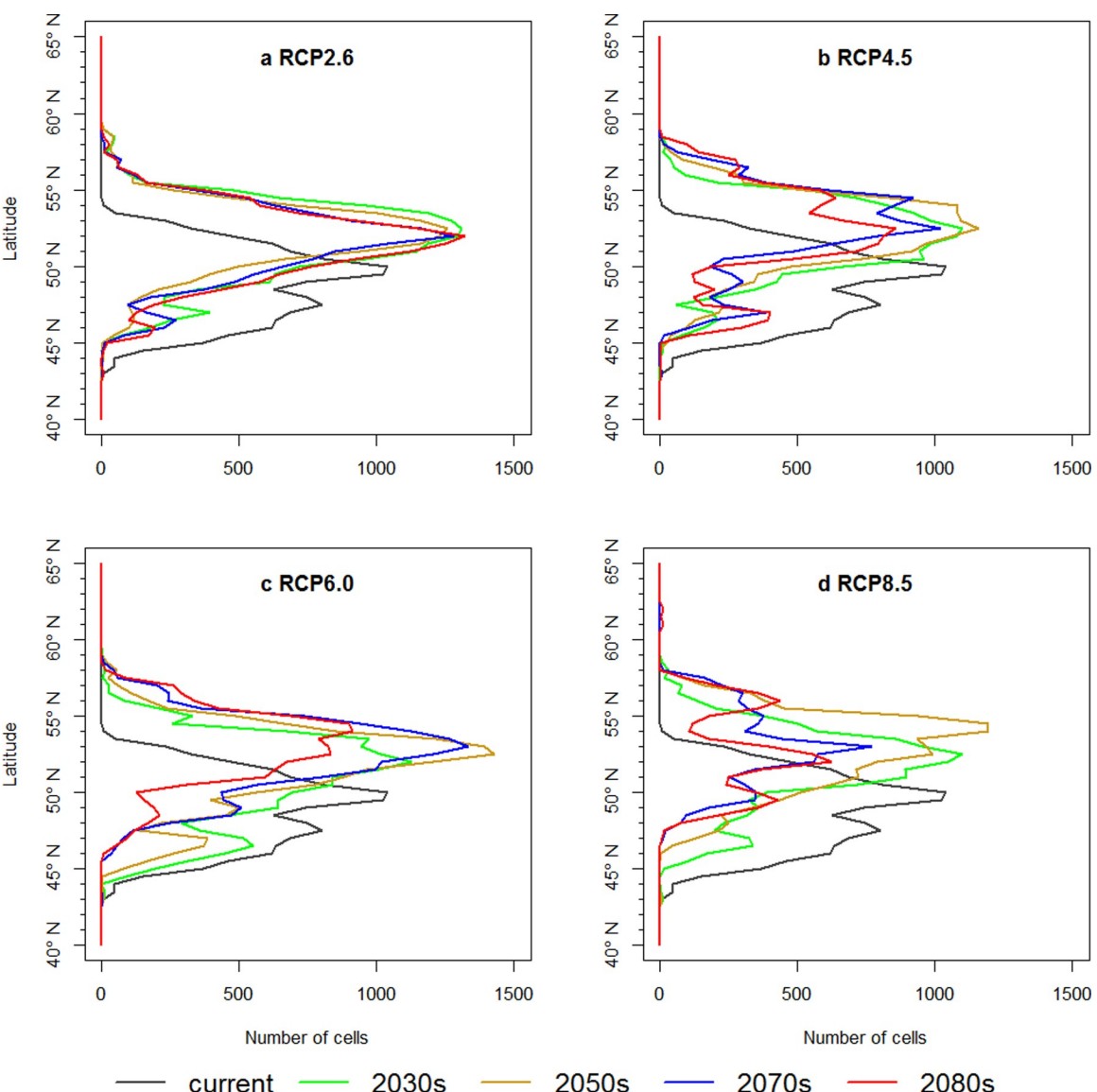

**Fig 2. Latitudinal pattern of predicted breeding range area (Bandwidth = 0.5 degree).**

higher than the number of predicted grids under current climate condition. However, under the high emission scenario—RCP8.5, it is slightly lower than current prediction. Our results suggest that the high emission scenario has more adverse effects on breeding range of the continental population of the red-crowned crane.

The proportion of predicted breeding area of the continental population of the red-crowned crane in China, decreases at the first and increases afterward under RCP2.6 and RCP4.5, and decreases continuously under RCP6.0 and RCP8.5 (Fig 4). But the proportion of grid cells under future climate is lower than that under current climate condition in China. The proportion of predicted grid cells in Russia, however, increases at first and decreases afterward under RCP2.6 and RCP4.5, and increases continuously under RCP6.0 and RCP8.5 (Fig 4). The proportion of grid cells under future climate is larger than that under current climate condition in Russia. Mongolia owns smallest proportion of grid cells of breeding area, and

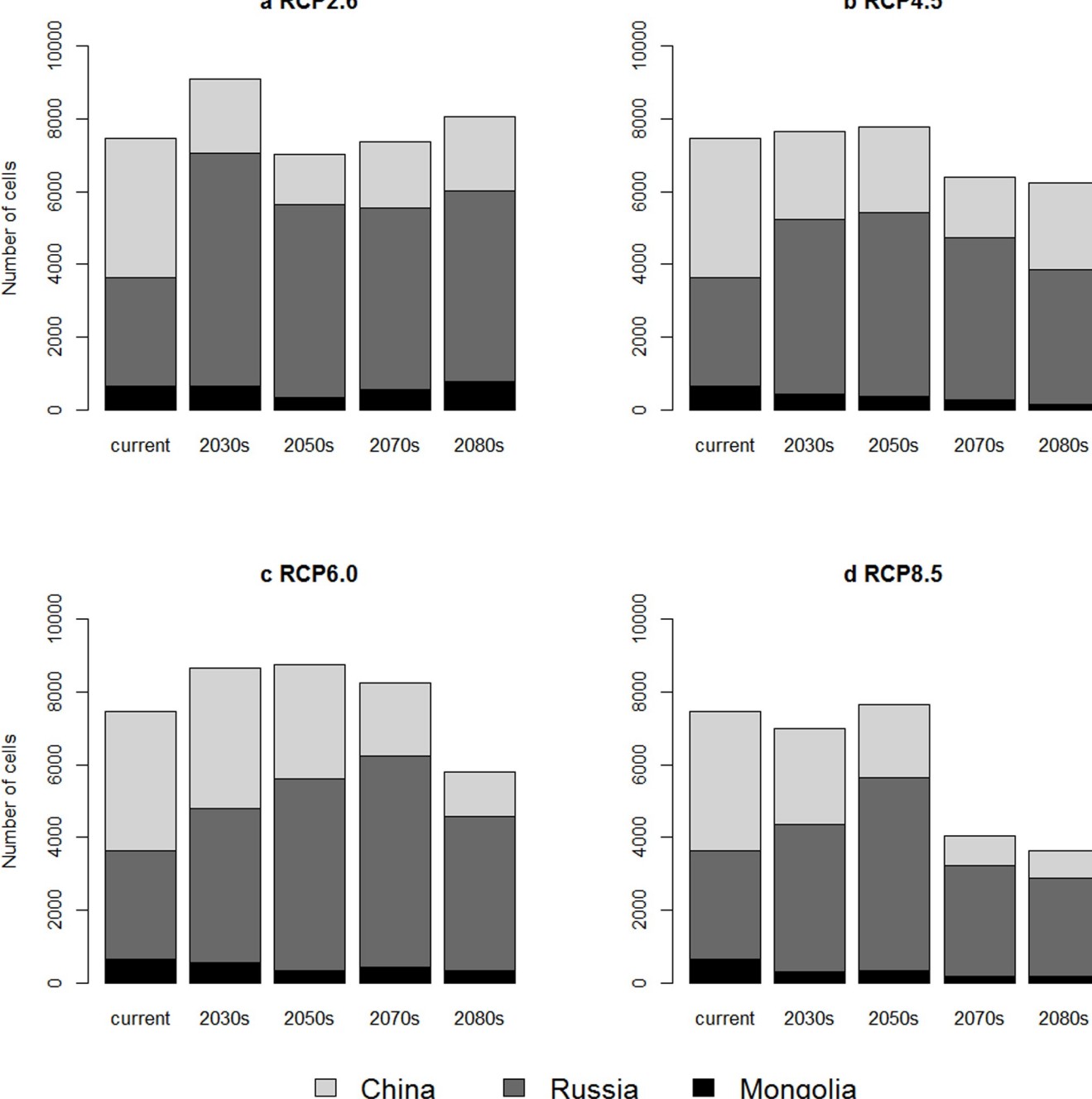

**Fig 3. The number of grid cells predicted to be breeding area of the continental population of the red-crowned crane in China, Russia and Mongolia.** a. low emission scenario–RCP2.6, b. intermediate scenario–RCP4.5, c. intermediate scenario–RCP6.0, d. high emission scenario–RCP8.5.

decreases under almost all emission scenarios. Our results revealed that China, which owns the largest breeding area currently, will lose most of its breeding area for the red-crowned crane. Russia, which has the second largest breeding area, will gain more breeding area in the face of climate change and would have the largest breeding area. It also means that Russia would have more responsibility to preserve this endangered species in this century.

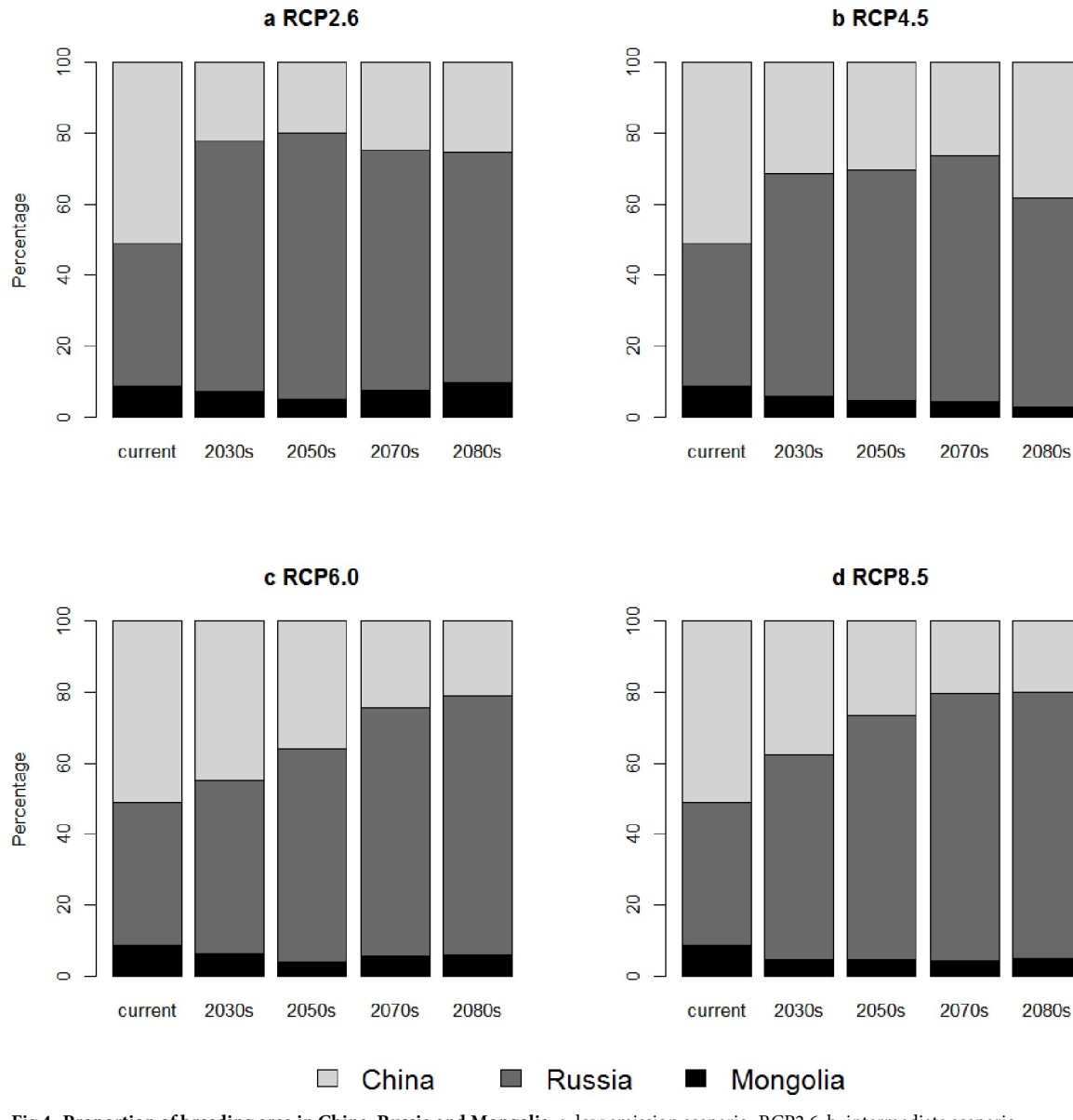

**Fig 4. Proportion of breeding area in China, Russia and Mongolia.** a. low emission scenario–RCP2.6, b. intermediate scenario–RCP4.5, c. intermediate scenario–RCP6.0, d. high emission scenario–RCP8.5.

We also calculated the proportion of grid cells predicted to be breeding area in the current actual breeding range of the red-crowned crane. The results show that the predicted grid cells suitable for breeding area in current actual breeding range from our model decrease very quickly in the near future under all climate change scenarios (Fig 5). Under the high emission scenario—RCP8.5 and the intermediate emission scenario—RCP6.0, the red-crowned crane would lose almost all of its actual breeding range at the end of this century. The situation is better under the low emission scenario—RCP2.6 and the intermediate emission scenario—RCP4.5. The continental population of the red-crowned crane would still have approximately 15% of its current actual breeding range at 2080s. At the center of this century, our model predicts the lowest area of suitable breeding area in current actual breeding range under RCP2.6 and RCP4.5. Under the intermediate emission scenario—RCP6.0, the red-crowned crane will

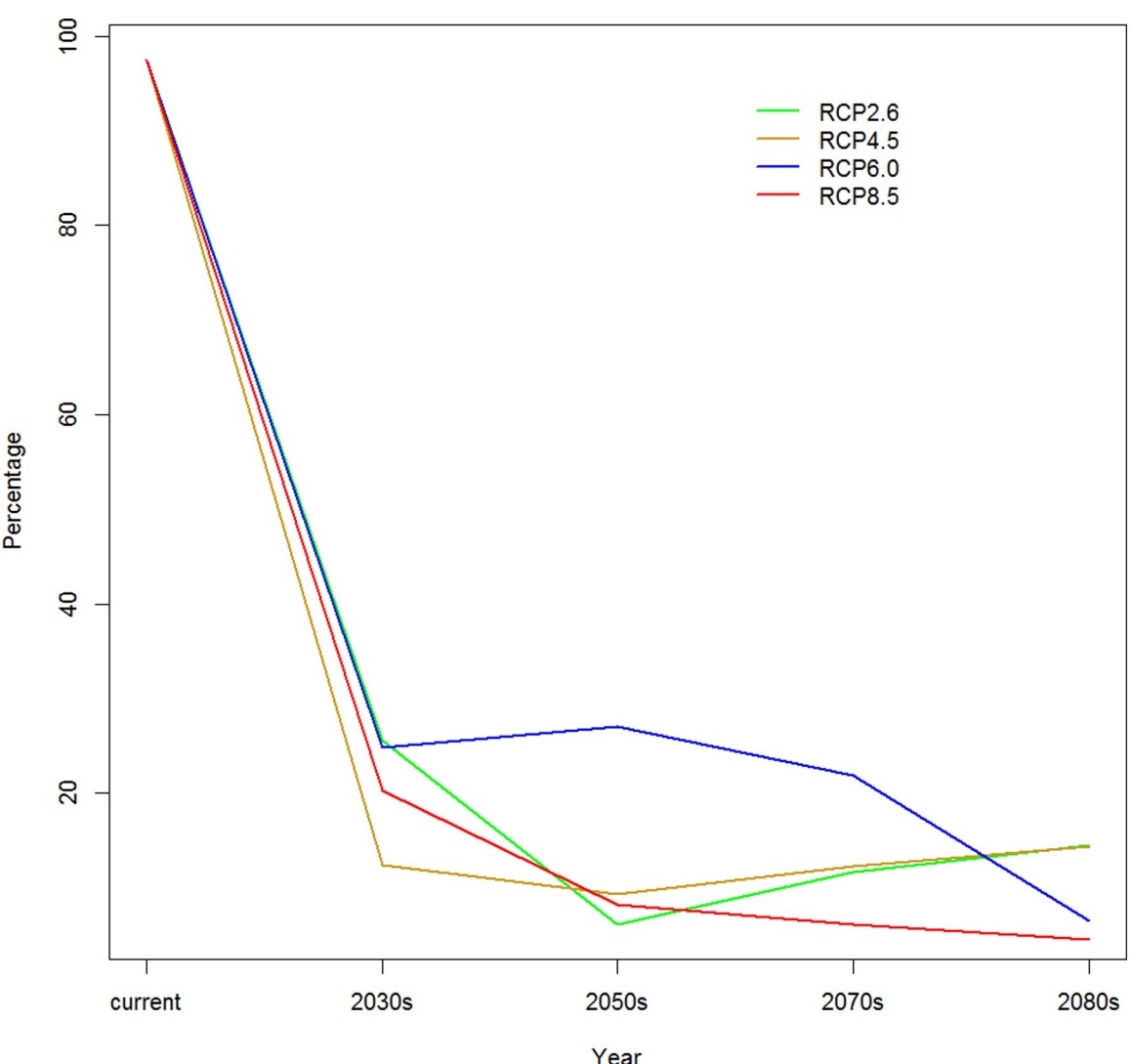

**Fig 5. Proportion of grid cells predicted to be breeding area in the actual breeding range.**

still have approximately 30% of its actual breeding range at the center of this century, which suggests that this emission scenario has less adverse effects on this species.

The average of relative importance of variable from our 10 cross-validation bioclimatic niche models shows that the most important variable is bio_15 (precipitation seasonality) (Fig 6). The AUC value decreases by approximately 30% after removing the effects of this variable in the models. The possible explanation of the highest importance of precipitation seasonality could be that this variable determines the vegetation cover patterns and food resources in this area. The second most important variable is Bio_4—temperature seasonality. The third most important variable contributing to bioclimatic niche models is Bio_13—precipitation of wettest month. The wettest month is the growing season for plant in the breeding range. It is also the breeding period for the red-crowned crane. Therefore the change of precipitation patterns and increasing temperature along with climate change would cause the shift of breeding range of this species.

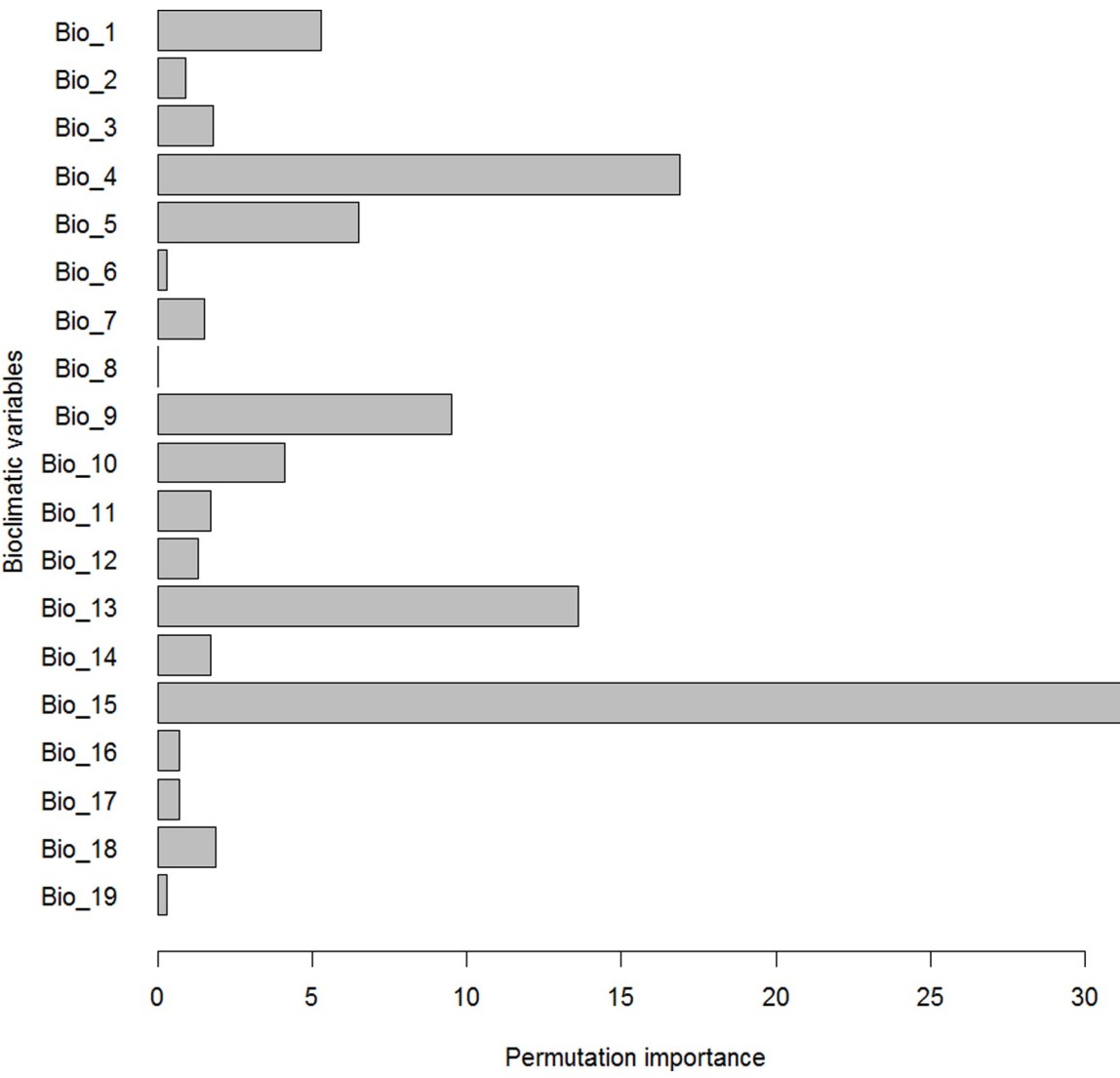

**Fig 6. Relative importance of variables in bioclimatic niche model.**

The mean value of human influence index in current actual breeding range of the continental population of the red-crowned crane is 14.4 (Fig 7). The average human influence index in predicted breeding area from our model is slightly higher than that of actual breeding range, which is 14.7. Our bioclimatic niche model predicts a much larger potential breeding area around the current actual breeding range. Because the Northeast China is one of the main grain production regions, and populated 0.107 billion people, the broader climatic-suitable breeding area predicted from our model may suffer from human activities such as urbanization, agriculture production, and habitat degradation. The center part of the Northeast China has many big cities and high population density. These factors restrict the red-crowned crane from inhabiting in all suitable breeding range under current climate condition.

As a result of climate change, the breeding area of the red-crowned crane will shift northward from current actual breeding range, where the population density is lower than that of current breeding area, and has less agricultural lands. Under the low emission scenario—RCP2.6, the intermediate emission scenario—RCP4.5, and the high emission scenario—

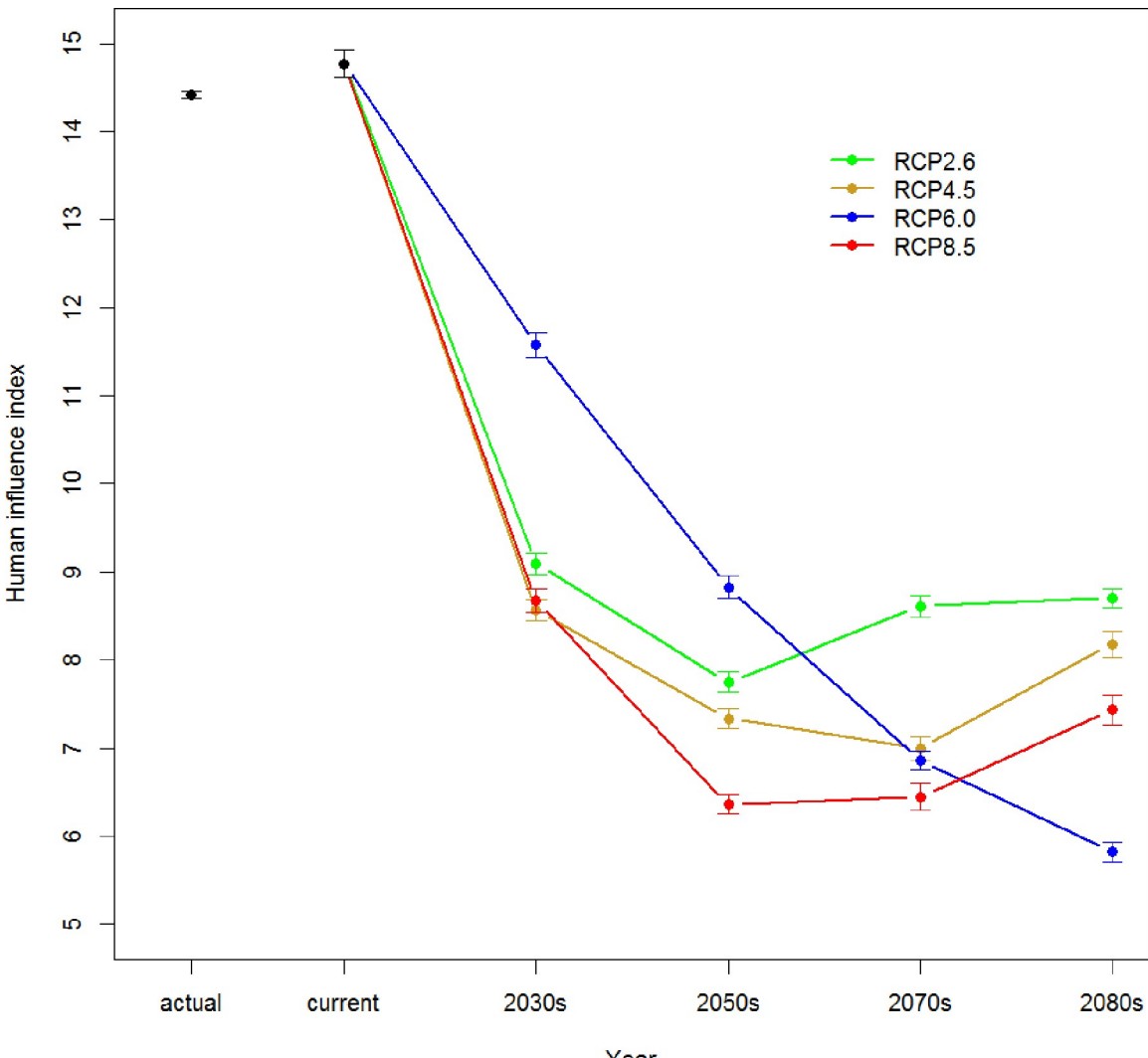

**Fig 7. Average human influence index in actual breeding area and predicted breeding area under climate change.** (Error bars show 95% confidence interval of average human influence index).

RCP8.5, the average of human influence index in the predicted breeding area maintains the similar pattern of change, decreasing quickly and then increasing slightly at the end of this century (Fig 7). The average human influence index under those three emission scenarios ranges from 7 to 9 at 2080s, much lower than the value in current actual breeding area. However, the average human influence index in breeding area under the intermediate emission scenario—RCP6.0 decreases continuously, and has the lowest value of 6 at the end of this century. This is because our model predicts that the breeding area will shift farther northward, and the main breeding area distributes around 54 arc-degree, where there is less human activity.

## Discussion

Our results demonstrated that bioclimatic niche modeling can predict the current breeding range of the continental population of the red-crowned crane very well. The excellent performance of the bioclimatic niche model makes us confident on the reliability of the prediction of future potential breeding range of red-crowned crane under changing climate. However, there are many other environmental factors such as food resource, nesting preference, land use, and

land cover, and biotic factors such as competition and predation restrict the red-crowned crane to fill all its climatic niche space. At large spatial scale, the climate is the most important factor shaping the breeding range of the red-crowned crane. Among 19 bioclimatic variables, the temperature seasonality (Bio4) and precipitation seasonality (Bio15) are the top two variables in determining the breeding range of the red-crowned crane. It suggests that the climate change, which results in spatial shift of seasonality of temperature and precipitation, would cause the shift of the breeding range.

There are many threats on the red-crowned crane in the breeding range, and affect the extent of breeding range. The population has been severely declined because of habitat modification, pollution and poaching [4,11,48]. Although the population was slightly increased after implementing the conservation policy made for this endangered species, and building protected area for it, such as Zhalong National Natural Reserve and Yancheng Biosphere Reserve, the population size is still very low, estimated at 2,750 [3,49]. Therefore it is under severe extinction risk from environmental change, especially the unprecedented climate change. Climate change would cause poleward shift of breeding range of the red-crowned crane. Some studies have already shown that some bird species extend their ranges northwards under climate change [23]. Climate change also affected the phenology of the red-crowned crane, which affected the adaptation capacity of this species [50,51]. Climate change also has indirect effects on the red-crowned crane, for example, one study showed that climate change altered the animal-plant interactions and thereby affected bird and plant communities [52]. The effects of climate change on the red-crowned crane and its breeding range, therefore, should be taken into consideration when making preservation policy for this endangered species.

The predicted shift of the breeding range of the red-crowned crane under changing climate from our bioclimatic niche models suggests that China would lose a proportion of the breeding range, Russia would gain some breeding range area. This effect might be overestimated, because the red-crowned crane has adaptation potential to climate change. Our model also neglected other abiotic and biotic factors affecting the breeding range of the red-crowned crane and the prediction accuracy of range shift in the face of climate change. This impact means that the role of conserving this endangered species would be changed, because the proportion of the breeding area in China, Russia and Mongolia would be changed. The area of breeding range in China would be shrunk as climate changes. The effectiveness of the protected area in China, therefore, would be jeopardized in the future. The increasing area of the breeding range in Russia and Mongolia means these two countries would take more responsibility to conserve this endangered species. However, the conservation policy in each country is different and does not necessarily consider the effects of climate change on the breeding range of this species. The international collaboration among the three countries would increase the conservation effectiveness of this species under climate change. Some studies showed that international collaboration can benefit the birds in Europe [53].

## Supporting information

**S1 Fig. Predicted breeding range of the continental population of the red-crowned crane at 2030s, 2050s, 2070s and 2080s.** Blue polygons are the current actual breeding area, green polygons are the predicted potential breeding area under current climate condition, and orange areas are the predicted potential breeding area climate change. Scenario is RCP4.5, intermediate emission scenario.
(DOC)

**S2 Fig. Predicted breeding range of the continental population of the red-crowned crane at 2030s, 2050s, 2070s and 2080s.** Blue polygons are the current actual breeding area, green

polygons are the predicted potential breeding area under current climate condition, and orange areas are the predicted potential breeding area climate change. Scenario is RCP6.0, intermediate emission scenario.
(DOC)

**S3 Fig. Predicted breeding range of the continental population of the red-crowned crane at 2030s, 2050s, 2070s and 2080s.** Blue polygons are the current actual breeding area, green polygons are the predicted potential breeding area under current climate condition, and orange areas are the predicted potential breeding area climate change. Scenario is RCP8.5, high emission scenario.
(DOC)

## Author Contributions

**Conceptualization:** Yinlong Zhang.

**Data curation:** Liwei Liu, Yongbo Wu.

**Formal analysis:** Jishan Liao.

**Funding acquisition:** Yinlong Zhang.

**Methodology:** Liwei Liu, Jishan Liao, Yongbo Wu.

**Project administration:** Yinlong Zhang.

**Software:** Jishan Liao.

**Supervision:** Yinlong Zhang.

**Validation:** Liwei Liu, Jishan Liao.

**Visualization:** Liwei Liu, Jishan Liao.

**Writing – original draft:** Liwei Liu, Jishan Liao.

**Writing – review & editing:** Liwei Liu, Jishan Liao, Yinlong Zhang.

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
