## [Decision Letter · Decision Letter 0]

16 Sep 2019

PONE-D-19-15791

Breeding range shift of the red-crowned crane (Grus japonensis) under climate change

PLOS ONE

Dear Mr Liao,

Thank you for submitting your manuscript to PLOS ONE. After careful consideration, we feel that it has merit but does not fully meet PLOS ONE’s publication criteria as it currently stands. Therefore, we invite you to submit a revised version of the manuscript that addresses the points raised during the review process.

We would appreciate receiving your revised manuscript by Oct 31 2019 11:59PM. To enhance the reproducibility of your results, we recommend that if applicable you deposit your laboratory protocols in protocols.io, where a protocol can be assigned its own identifier (DOI) such that it can be cited independently in the future. For instructions see: http://journals.plos.org/plosone/s/submission-guidelines#loc-laboratory-protocols

We look forward to receiving your revised manuscript.

Kind regards,

Bi-Song Yue, Ph.D

Academic Editor

PLOS ONE

Journal Requirements:

This project was jointly supported by National Key R&D Program of China（2016YFC0502704）

4. Please amend the manuscript submission data (via Edit Submission) to include author Yinlong Zhang.

Reviewers' comments:

Reviewer's Responses to Questions

**Comments to the Author**

1. Is the manuscript technically sound, and do the data support the conclusions?

Reviewer #1: Yes

Reviewer #2: Yes

Reviewer #3: Partly

2. Has the statistical analysis been performed appropriately and rigorously? 

Reviewer #1: Yes

Reviewer #2: No

Reviewer #3: Yes

3. Have the authors made all data underlying the findings in their manuscript fully available?

Reviewer #1: Yes

Reviewer #2: Yes

Reviewer #3: Yes

4. Is the manuscript presented in an intelligible fashion and written in standard English?

Reviewer #1: Yes

Reviewer #2: Yes

Reviewer #3: No

5. Review Comments to the Author

Reviewer #1: Liu, Liao, Wu, and Zhang present an interesting and important study on how the breeding range of the red-crowned crane could shift under scenarios of future climate. I find that the manuscript is good overall and should be suitable for publication pending minor revision. The main points that I would like to see addressed follow:

1) Provide a clearer statement on which statistical downscale method is applied to future climate fields (I assume the delta method), and that the downscale is a derived product from the CCAFS GCM Data Portal (unless you ran the statistical downscaling?).

2) Provide some justification for choosing HadGEM2-ES instead of a multi-model ensemble mean, or some other individual model (e.g., CCSM4, CESM1). One or two sentences should suffice, ideally with a citation for a study that finds strong realism HadGEM2-ES.

3) Point 2 alludes to the caveat that different models produce different results. That said, it would be helpful to show the reader additional supplementary figures -- temperature and precipitation difference maps for HadGEM2-ES minus baseline modern climate for each of the RCPs (2.6, 4.5, 6.0, 8.5) at year 2100 CE. Annual anomalies will suffice (4 temperature anomaly and 4 precipitation anomaly maps). This will at least give the reader a visual depiction of how HadGEM2-ES projects future climate. I suspect the anomaly patterns will reinforce the importance of precipitation as mentioned in the results.

4) Please also see minor edits in the attached PDF.

Reviewer #2: The topic of how climate change may affect future potential distributions of species is a very important one, and the authors do well to explore this question for one of the 11 endangered species of cranes. Such an analyses has great value especially given the relatively large distribution of this particular species.

The questions I have relate entire to the modeling technique, and authors require to respond to all of these, and perhaps re-run some analyses to validate some of the responses.

While MAXENT is a great tool for this kind of work, there is a lot of emerging work that shows the default setting of the software to not be optimal. Since the settings and software as not fully explanatory, using MAXENT is in a sense using a blackbox with limited insight into how the final results are achieved. A vast majority of the evaluations make the point that independent models that were more parsimonious usually work much better than the more complex models that are run in MAXENT using a large number of variables.

Also, not being able to use spatially filtered records in this particular case has the potential of violating the model assumptions of representative sampling. Can the authors provide a sound explanation and reasoning for their methodology?

One final, and relatively minor comment is the use of variables many of which are known to be auto-correlated spatially. How do the authors think variable redundancy is affecting their outputs, and why should the final models presented in this manuscript be thought to be robust to variable redundancy?

Reviewer #3: This is an interesting and important study on a timely subject but the manuscript suffers from multiple factual errors, sub-standard writing, and a lack of sufficient, up-to-date citations. The authors’ bioclimatic niche models predict a northward shift of the breeding range of the red-crowned crane under climate change, from China into Russia. They point out that national conservation policies in China and Russia as well as Mongolia differ and currently do not consider the effects of climate change, and that the effects of climate change on the red-crowned crane, therefore, should be taken into consideration in conservation planning and international cooperation.

The range shift modeling appears sound to me, but should be validated by an expert in the modeling methods they used. Once these issues have been addressed, I would recommend publication in PLOS ONE because and the results are important and deserving of a wide audience. However, in order to be eligible for publication in PLOS ONE, the factual errors throughout must be corrected, new or updated citations should be added, and the body text thoroughly revised. In its current form, the manuscript is difficult to read and suffers from outdated or erroneous statements throughout that undermine its credibility.

Below I provide a partial list of some of the factual errors and discrepancies and suggested changes and updated citations. Currently the authors list 26 citations for the paper; this list should be augmented through a thorough, up-to-date use of authoritative scientific literature. Furthermore, I strongly recommend the manuscript be rewritten from beginning to end, ideally with assistance from a native or fluent English speaker, to correct many issues with incorrect grammar and unclear or confusing wording.

If the problems with errors, citations, and language are addressed I believe their findings should be considered for the PLOS ONE homepage because they represent significant impacts to biodiversity from human activity that should be highlighted. Specifically, range shifts under changing climate will necessitate changes to conservation planning and increased international collaboration for this and other endangered species threatened by climate change. As mentioned above, selected specific problems and suggestions appear below. Thank you for the opportunity to review this paper. I would be happy to review a revised draft.

>>

In line 51, the authors write “The red-crowned crane (Grus japonensis) is the largest and rarest crane in the world.” However, the authors do not provide a citation and according to the Handbook of Birds of the World, this species does not appear to be the tallest crane; both the Sarus Crane and Whooping Crane reach taller maximum heights (176 cm and 160 cm compared to Red-crowned Crane’s 152 cm).

This species is also not currently the rarest crane; the Whooping Crane is rarer with population of fewer than 600 individuals compared to Red-crowned Crane’s estimated 3000 individuals.

Archibald, G.W., Meine, C.D. & Garcia, E.F.J. (2019). Red-crowned Crane (Grus japonensis). In: del Hoyo, J., Elliott, A., Sargatal, J., Christie, D.A. & de Juana, E. (eds.). Handbook of the Birds of the World Alive. Lynx Edicions, Barcelona: https://www.hbw.com/node/53564

Archibald, G.W., Meine, C.D. & Garcia, E.F.J. (2019). Sarus Crane (Antigone antigone). In: del Hoyo, J., Elliott, A., Sargatal, J., Christie, D.A. & de Juana, E. (eds.). Handbook of the Birds of the World Alive. Lynx Edicions, Barcelona: https://www.hbw.com/node/53557

Archibald, G.W., Meine, C.D. & Garcia, E.F.J. (2019). Whooping Crane (Grus americana). In: del Hoyo, J., Elliott, A., Sargatal, J., Christie, D.A. & de Juana, E. (eds.). Handbook of the Birds of the World Alive. Lynx Edicions, Barcelona: https://www.hbw.com/node/53562

Authors should consult HBW or other up-to-date, authoritative sources to back up their claims. They might rephrase their opening sentence as follows: “The red-crowned crane (Grus japonensis) is among the largest and rarest cranes in the world.”

In lines 54-55, the authors state “the range of this species contracted and its population declined greatly over last centuries, especially in recent years.” However, according to Archibald et al. (2019; cited in full above) their numbers were lower in the middle of the last century than in recent years, reaching lowest point in the 1950’s.

In line 57, the authors state the species is critical[ly] endangered as listed by the IUCN, citing the Russian Red Book; the authors should instead cite the IUCN authority directly, which is BirdLife International, where the latest listing shows this species as Endangered, not Critically Endangered: BirdLife International 2016. Grus japonensis. The IUCN Red List of Threatened Species 2016: e.T22692167A93339099. http://dx.doi.org/10.2305/IUCN.UK.2016-3.RLTS.T22692167A93339099.en

In lines 72-74, the authors write: “The anthropogenic habitat change, such as development of agriculture, urbanization, and conversion of wetland to human-built area, had caused great decline of the population and loss of habitats of the red-crowned crane in recent years.” The citation they provide, while authoritative, is from 1983; the authors should add to this citation with a more recent authoritative one. For example, Archibald et al. (2019), who write, “Loss and degradation of wetlands due to agricultural and industrial development constitute main threat to the species’ breeding areas… in the Sanjiang Plain in NE China, and Amur Basin in... SE Russia.”

Archibald et al. (2019) also cite Zhou et al. (2016), who suggest that wild cranes are harvested to supplement the captive-bred population in China, and that this activity is driving population declines:

Zhou, D. et al. (2016). A growing captive population erodes the wild Red-crowned Cranes (Grus japonensis) in China. Avian Research 7: 22

In line 76, the authors state "This species prefers to nesting in wetlands and rivers.”According to Archibald et al. (2019), however, this species nests in wetlands (including “reed, sedge and cat-tail marshes, and extensive bogs and wet meadows”) but not rivers, with nests built in relatively deep standing water. The authors should provide a citation for their statements on crane nesting ecology and I suggest removing the reference to nesting in rivers unless the authors can show clear documentation of this.

Likewise, in lines 78-79 the authors write “In the wintering range, the red-crowned crane prefers to inhabit in paddy fields, grassy tidal flats, and mudflats…” but do not mention that this species is known to winter at rivers, as described by Archibald et al. (2019). Again, I suggest checking their habitat statements against updated authoritative sources.

In lines 170-171, the authors state, “We used Maximum Entropy (MaxEnt) [10], which is the most popular and powerful modeling technique for predicting distribution of species [19].”Because there is no universal measure of “the most population and powerful modeling technique…”authors would better revise this to state to “a powerful, novel modeling technique…”

Later, in lines 420-421, the authors state “The population has been severely declined because of habitat modification, pollution and poaching.” The authors should provide citations for this statement. Habitat modification as a threat is supported by Archibald et al. (2019) but other references could be named; the authors should cite evidence of pollution as a threat to red-crowned cranes as well. Zhou et al. (2016), mentioned above, might be used to support the poaching statement; are there other publications that suggest or demonstrate poaching of red-crowned cranes? Archibald et al. (2019) mention other threats to this species include "overharvesting of wetland resources, human disturbance, intentional setting of fires in breeding areas, and poisoning."

In lines 422-424, the authors state, “Although the population was slightly increased after implementing the conservation policy made for this critical endangered species, and building protected area for it… such as Zhalong National Natural Reserve and Yancheng Biosphere Reserve…” Again, the authors need to back up such statements with evidence as provided in citations, and again, this species is not considered critically endangered but endangered by the IUCN. What conservation policies resulted in increases of this species and how was this measured? Where the protected areas mentioned created specifically for red-crowned cranes or were they one of several or many reasons for their creation?

In lines 451-454, the authors state, “The international collaboration among the three countries would increase the conservation effectiveness of this species, for instance, assisted movement of the red-crowned crane from China to Mongolia and Russia…”What do the authors mean by “assisted movement”? Translocation? I recommend taking such suggestions out of this paper unless there is well-documented support for them.

6. PLOS authors have the option to publish the peer review history of their article (what does this mean?). If published, this will include your full peer review and any attached files.

Reviewer #1: No

Reviewer #2: No

Reviewer #3: No

---

## [Author Response · Author response to Decision Letter 0]

17 Feb 2020

Reviewer #1: Liu, Liao, Wu, and Zhang present an interesting and important study on how the breeding range of the red-crowned crane could shift under scenarios of future climate. I find that the manuscript is good overall and should be suitable for publication pending minor revision. The main points that I would like to see addressed follow:

1) Provide a clearer statement on which statistical downscale method is applied to future climate fields (I assume the delta method), and that the downscale is a derived product from the CCAFS GCM Data Portal (unless you ran the statistical downscaling?).

We used the downscaled CMIP5 climate data based on delta method. We downloaded the data from http://www.ccafs-climate.org/data_spatial_downscaling/. The delta method was described in http://www.ccafs-climate.org/statistical_downscaling_delta_cmip5/

2) Provide some justification for choosing HadGEM2-ES instead of a multi-model ensemble mean, or some other individual model (e.g., CCSM4, CESM1). One or two sentences should suffice, ideally with a citation for a study that finds strong realism HadGEM2-ES.

There are about 28 GCMs available from http://www.ccafs-climate.org/statistical_downscaling_delta_cmip5/. We included all four RCPs and future time periods in our study. We also run 10-fold cross validation of MaxEnt. It means that we had to generated about 28*4*4*10 = 4480 prediction rasters if we chose multi-model ensemble mean. It’s very time cosuming. So we choose HadGEM2-ES, which was prove to perform well in previous studies:

1. Evaluation by prediction of the natural range shrinkage of Quercus ilex L. in eastern Algeria 

2. Protecting rare and endangered species under climate change on the Qinghai Plateau, China

3. Species-specific ecological niche modelling predicts different range contractions for Lutzomyia intermedia and a related vector of Leishmania braziliensis following climate change in South America

4. Combining Niche Modelling, Land-Use Change, and Genetic Information to Assess the Conservation Status of Pouteria splendens Populations in Central Chile

5. The impact of climate change on the geographical distribution of two vectors of Chagas disease: implications for the force of infection

3) Point 2 alludes to the caveat that different models produce different results. That said, it would be helpful to show the reader additional supplementary figures -- temperature and precipitation difference maps for HadGEM2-ES minus baseline modern climate for each of the RCPs (2.6, 4.5, 6.0, 8.5) at year 2100 CE. Annual anomalies will suffice (4 temperature anomaly and 4 precipitation anomaly maps). This will at least give the reader a visual depiction of how HadGEM2-ES projects future climate. I suspect the anomaly patterns will reinforce the importance of precipitation as mentioned in the results.

We produced precipitation and temperature change maps.

4) Please also see minor edits in the attached PDF.

Revised accordingly.

Reviewer #2: The topic of how climate change may affect future potential distributions of species is a very important one, and the authors do well to explore this question for one of the 11 endangered species of cranes. Such an analyses has great value especially given the relatively large distribution of this particular species.

The questions I have relate entire to the modeling technique, and authors require to respond to all of these, and perhaps re-run some analyses to validate some of the responses.

While MAXENT is a great tool for this kind of work, there is a lot of emerging work that shows the default setting of the software to not be optimal. Since the settings and software as not fully explanatory, using MAXENT is in a sense using a blackbox with limited insight into how the final results are achieved. A vast majority of the evaluations make the point that independent models that were more parsimonious usually work much better than the more complex models that are run in MAXENT using a large number of variables.

The author of MaxEnt suggested using default settings. “The use of default settings is justified provided that they have been validated over a wide range of species, environmental conditions, numbers of occurrences, and amounts of sample selection bias” as stated in their paper “Modeling of species distributions with Maxent: new extensions”. We admit that different combination of parameters would produce different results, and may get higher model performance index values. The spatial patterns of predictions would not be changed very much and will not change our main results and conclusions.

Also, not being able to use spatially filtered records in this particular case has the potential of violating the model assumptions of representative sampling. Can the authors provide a sound explanation and reasoning for their methodology?

We downloaded occurrence points from GBIF. We also searched occurrence points from other online resources and published literature. We found very limited number of occurrence points in the breeding areas. It’s not enough to build niche model. We compared the occurrence points with extent of occurrence polygon used in our study. We found almost all of them were located in the polygons. This amount of occurrence points cannot represent the niche of this species. So we chose the extent of occurrence polygon over the reported occurrence points.

One final, and relatively minor comment is the use of variables many of which are known to be auto-correlated spatially. How do the authors think variable redundancy is affecting their outputs, and why should the final models presented in this manuscript be thought to be robust to variable redundancy?

We admit that collinearity among predictors is a very important issue in modeling. But for MaxEnt, “there is less need to remove correlated variables (unless some of them are known to be ecologically irrelevant)” as stated in MaxEnt author’s paper: A statistical explanation of MaxEnt for ecologists. 

This model implicitly incorporated the percentage contribution of each variable to the final solution. For a pair of highly-correlated variables, one variable contributes more to the final model, where the other variable contributes negligibly to the model. We validated this by calculating the correlation matrix of 19 bioclimatic variables and relative importance of variables in MaxEnt. Another paper titled “Species distribution modelling using bioclimatic variables to determine the impacts of a changing climate on the western ringtail possum (Pseudocheirus occidentals; Pseudocheiridae)” also found same results.

Reviewer #3: This is an interesting and important study on a timely subject but the manuscript suffers from multiple factual errors, sub-standard writing, and a lack of sufficient, up-to-date citations. The authors’ bioclimatic niche models predict a northward shift of the breeding range of the red-crowned crane under climate change, from China into Russia. They point out that national conservation policies in China and Russia as well as Mongolia differ and currently do not consider the effects of climate change, and that the effects of climate change on the red-crowned crane, therefore, should be taken into consideration in conservation planning and international cooperation.

The range shift modeling appears sound to me, but should be validated by an expert in the modeling methods they used. Once these issues have been addressed, I would recommend publication in PLOS ONE because and the results are important and deserving of a wide audience. However, in order to be eligible for publication in PLOS ONE, the factual errors throughout must be corrected, new or updated citations should be added, and the body text thoroughly revised. In its current form, the manuscript is difficult to read and suffers from outdated or erroneous statements throughout that undermine its credibility.

Below I provide a partial list of some of the factual errors and discrepancies and suggested changes and updated citations. Currently the authors list 26 citations for the paper; this list should be augmented through a thorough, up-to-date use of authoritative scientific literature. Furthermore, I strongly recommend the manuscript be rewritten from beginning to end, ideally with assistance from a native or fluent English speaker, to correct many issues with incorrect grammar and unclear or confusing wording.

We added more citations after literature review. The number of citations become 53 now.

If the problems with errors, citations, and language are addressed I believe their findings should be considered for the PLOS ONE homepage because they represent significant impacts to biodiversity from human activity that should be highlighted. Specifically, range shifts under changing climate will necessitate changes to conservation planning and increased international collaboration for this and other endangered species threatened by climate change. As mentioned above, selected specific problems and suggestions appear below. Thank you for the opportunity to review this paper. I would be happy to review a revised draft.

>>

In line 51, the authors write “The red-crowned crane (Grus japonensis) is the largest and rarest crane in the world.” However, the authors do not provide a citation and according to the Handbook of Birds of the World, this species does not appear to be the tallest crane; both the Sarus Crane and Whooping Crane reach taller maximum heights (176 cm and 160 cm compared to Red-crowned Crane’s 152 cm).

This species is also not currently the rarest crane; the Whooping Crane is rarer with population of fewer than 600 individuals compared to Red-crowned Crane’s estimated 3000 individuals.

Archibald, G.W., Meine, C.D. & Garcia, E.F.J. (2019). Red-crowned Crane (Grus japonensis). In: del Hoyo, J., Elliott, A., Sargatal, J., Christie, D.A. & de Juana, E. (eds.). Handbook of the Birds of the World Alive. Lynx Edicions, Barcelona: https://www.hbw.com/node/53564

Archibald, G.W., Meine, C.D. & Garcia, E.F.J. (2019). Sarus Crane (Antigone antigone). In: del Hoyo, J., Elliott, A., Sargatal, J., Christie, D.A. & de Juana, E. (eds.). Handbook of the Birds of the World Alive. Lynx Edicions, Barcelona: https://www.hbw.com/node/53557

Archibald, G.W., Meine, C.D. & Garcia, E.F.J. (2019). Whooping Crane (Grus americana). In: del Hoyo, J., Elliott, A., Sargatal, J., Christie, D.A. & de Juana, E. (eds.). Handbook of the Birds of the World Alive. Lynx Edicions, Barcelona: https://www.hbw.com/node/53562

Authors should consult HBW or other up-to-date, authoritative sources to back up their claims. They might rephrase their opening sentence as follows: “The red-crowned crane (Grus japonensis) is among the largest and rarest cranes in the world.”

Revised accordingly.

In lines 54-55, the authors state “the range of this species contracted and its population declined greatly over last centuries, especially in recent years.” However, according to Archibald et al. (2019; cited in full above) their numbers were lower in the middle of the last century than in recent years, reaching lowest point in the 1950’s.

In line 57, the authors state the species is critical[ly] endangered as listed by the IUCN, citing the Russian Red Book; the authors should instead cite the IUCN authority directly, which is BirdLife International, where the latest listing shows this species as Endangered, not Critically Endangered: 

BirdLife International 2016. Grus japonensis. The IUCN Red List of Threatened Species 2016: e.T22692167A93339099. http://dx.doi.org/10.2305/IUCN.UK.2016-3.RLTS.T22692167A93339099.en

In lines 72-74, the authors write: “The anthropogenic habitat change, such as development of agriculture, urbanization, and conversion of wetland to human-built area, had caused great decline of the population and loss of habitats of the red-crowned crane in recent years.” The citation they provide, while authoritative, is from 1983; the authors should add to this citation with a more recent authoritative one. For example, Archibald et al. (2019), who write, “Loss and degradation of wetlands due to agricultural and industrial development constitute main threat to the species’ breeding areas… in the Sanjiang Plain in NE China, and Amur Basin in... SE Russia.”

We also added another two citations:

Ma Z, Wang Z, Tang H. History of Red-crowned Crane Grus japonensis and its habitats in China. Bird Conserv Int. 1998; doi:10.1017/S0959270900003592

Claire M. Mirande; James T. Harris. Crane Conservation Strategy. Baraboo, Wisconsin, USA: Minuteman Press; 2019.

Archibald et al. (2019) also cite Zhou et al. (2016), who suggest that wild cranes are harvested to supplement the captive-bred population in China, and that this activity is driving population declines: Zhou, D. et al. (2016). A growing captive population erodes the wild Red-crowned Cranes (Grus japonensis) in China. Avian Research 7: 22

We added the recommended citation and changed the sentence to “The anthropogenic habitat change, such as development of agriculture, urbanization, and conversion of wetland to human-built area, and increasing of captive red-crowned cranes harvested from the wild in China, have caused a significant decline of the population and loss of habitats of the red-crowned crane”.

In line 76, the authors state "This species prefers to nesting in wetlands and rivers.”According to Archibald et al. (2019), however, this species nests in wetlands (including “reed, sedge and cat-tail marshes, and extensive bogs and wet meadows”) but not rivers, with nests built in relatively deep standing water. The authors should provide a citation for their statements on crane nesting ecology and I suggest removing the reference to nesting in rivers unless the authors can show clear documentation of this.

We changed the sentence to “This species prefers to nesting in wetlands and river floodplains.” We also added three relevant references,

Johnsgard PA. Cranes of the World: Japanese Crane (Grus japonensis). Cranes of the World, by Paul Johnsgard. 1983. p. 21. 

Curt D. Meine; George W. Archibald. The cranes : status survey and conservation action plan. IUCN, Gland, Switzerland, and Cambridge, U.K.; 1996.

Heim W, Trense D, Sokolova G V., Kitagawa T. Increased Populations of Endangered Cranes After Amur River Flood. Waterbirds. 2017; doi:10.1675/063.040.0309

Likewise, in lines 78-79 the authors write “In the wintering range, the red-crowned crane prefers to inhabit in paddy fields, grassy tidal flats, and mudflats…” but do not mention that this species is known to winter at rivers, as described by Archibald et al. (2019). Again, I suggest checking their habitat statements against updated authoritative sources.

We added two relevant references,

SU L, ZOU H. Status, threats and conservation needs for the continental population of the Red-crowned Crane. Chinese Birds. 2012;3: 147–164. doi:10.5122/cbirds.2012.0030

Liu C, Jiang H, Zhang S, Li C, Hou Y, Qian F. Multi-scale analysis to uncover habitat use of red-crowned cranes: Implications for conservation. Curr Zool. 2013;59: 604–617. doi:10.1093/czoolo/59.5.604

In lines 170-171, the authors state, “We used Maximum Entropy (MaxEnt) [10], which is the most popular and powerful modeling technique for predicting distribution of species [19].”Because there is no universal measure of “the most population and powerful modeling technique…”authors would better revise this to state to “a powerful, novel modeling technique…”

We changed the sentence to “For example, Wenjun et al. used Maximum Entropy (MaxEnt) [35], a powerful and popular ecological niche modeling technique, and a variety of variables to predict the effects of climate change on the breeding range of this species in China [7].”

Later, in lines 420-421, the authors state “The population has been severely declined because of habitat modification, pollution and poaching.” The authors should provide citations for this statement. Habitat modification as a threat is supported by Archibald et al. (2019) but other references could be named; the authors should cite evidence of pollution as a threat to red-crowned cranes as well. Zhou et al. (2016), mentioned above, might be used to support the poaching statement; are there other publications that suggest or demonstrate poaching of red-crowned cranes? Archibald et al. (2019) mention other threats to this species include "overharvesting of wetland resources, human disturbance, intentional setting of fires in breeding areas, and poisoning."

Revised accordingly. 

We also added another reference,

Sun K-J, Hijikata N, Ichinose T, Higuchi H. The Migration Flyways and Protection of Cranes in China. Global Environmental Research ©2015 AIRIES. 2015.

In lines 422-424, the authors state, “Although the population was slightly increased after implementing the conservation policy made for this critical endangered species, and building protected area for it… such as Zhalong National Natural Reserve and Yancheng Biosphere Reserve…” Again, the authors need to back up such statements with evidence as provided in citations, and again, this species is not considered critically endangered but endangered by the IUCN. What conservation policies resulted in increases of this species and how was this measured? Where the protected areas mentioned created specifically for red-crowned cranes or were they one of several or many reasons for their creation?

Zhalong National Natural Reserve was created mainly for cranes. It is the largest breeding ground for the Red-crowned cranes. There are about 1/5 of them in the world living in Zhalong. 

We also added a reference which studied the population dynamics of red-crowned crane in the bread areas.

 Fawen QIAN Guohai YU, Youzhong YU, Jun YANG, Siliang PANG, Renzu PIAO HJ. Survey of breeding populations of the Red-Crowned Crane (Grus japonensis) in the Songnen Plain, northeastern China. Chinese Birds. pp. 217–224.

In lines 451-454, the authors state, “The international collaboration among the three countries would increase the conservation effectiveness of this species, for instance, assisted movement of the red-crowned crane from China to Mongolia and Russia…”What do the authors mean by “assisted movement”? Translocation? I recommend taking such suggestions out of this paper unless there is well-documented support for them.

We deleted “for instance, assisted movement of the red-crowned crane from China to Mongolia and Russia”.

---

## [Editor Report · Decision Letter 1]

20 Feb 2020

Breeding range shift of the red-crowned crane (Grus japonensis) under climate change

PONE-D-19-15791R1

Dear Dr. Zhang,

We are pleased to inform you that your manuscript has been judged scientifically suitable for publication and will be formally accepted for publication once it complies with all outstanding technical requirements.

With kind regards,

Bi-Song Yue, Ph.D

Academic Editor

PLOS ONE

---

## [Editor Report · Acceptance letter]

24 Feb 2020

PONE-D-19-15791R1 

Breeding range shift of the red-crowned crane (*Grus japonensis*) under climate change 

Dear Dr. Zhang:

I am pleased to inform you that your manuscript has been deemed suitable for publication in PLOS ONE. Congratulations! Your manuscript is now with our production department. 

With kind regards,

on behalf of

Dr. Bi-Song Yue 

Academic Editor

PLOS ONE